# Electrochemical Impedance Investigation of Dye-Sensitized Solar Cells Based on Electrospun TiO_2_ Nanofibers Photoanodes

**DOI:** 10.3390/ma15176175

**Published:** 2022-09-05

**Authors:** Hany M. Abd El-Lateef, Mai M. Khalaf, Van-Duong Dao, Ibrahim M. A. Mohamed

**Affiliations:** 1Department of Chemistry, College of Science, King Faisal University, Al-Ahsa 31982, Saudi Arabia; 2Department of Chemistry, Faculty of Science, Sohag University, Sohag 82524, Egypt; 3Faculty of Biotechnology, Chemistry and Environmental Engineering, Phenikaa University, Hanoi 10000, Vietnam

**Keywords:** impedance analysis, N-doping, photoanode substrate, solar energy, nanofibers

## Abstract

This work investigates an electrochemical impedance analysis based on synthesized TiO_2_ nanofibers (NFs) photoanodes, which were fabricated via electrospinning and calcination. The investigated photoanode substrate NFs were studied in terms of physicochemical tools to investigate their morphological character, crystallinity, and chemical contents via scanning electron microscope (SEM), X-ray photoelectron spectroscopy (XPS), and X-ray diffraction (XRD) analyses. As a result, the studied photoanode substrate NFs were applied to fabricate dye-sensitized solar cells (DSCs), and the electrochemical impedance analysis (EIS) was studied in terms of equivalent circuit fitting and impacts of N-doping, the latter of which was approved via XPS analysis. N-doping has a considerable role in the enhancement of charge transfers, which could be due to the strong interactions between active-site N atoms and the used photosensitizer.

## 1. Introduction

Electrochemical impedance spectroscopy (EIS) has become an outstanding technique to understand the charge transfer processes of electrochemical devices [1,2], such as fuel cells [3], Li-based batteries [4], perovskite solar cells [5], solid-state Batteries [6], and dye-sensitized solar cells (DSCs) [7]. An analysis and understanding of EIS can provide key information about the investigated electrochemical device, especially about electron lifetime and charge recombination at both anodes and cathodes [8]. An equivalent circuit analysis of the extracted EIS data could estimate series resistances (*R*_s_) and charge-transfer resistance (*R*_ct_), in addition to charge recombination or trap distribution parameters [8,9,10]. Therefore, this work studied an EIS analysis of DSCs based on electrospun nanofibers (NFs) attached with photosensitizers as a photoanode.

The energy crisis provides a high motivation for scientific society to find an economically renewable alternative instead of simple fossil fuels, which have a negative impact on the environment. The best alternative from the view of the environment is renewable solar energy, which has a renewable character in addition to being environmentally safe [11]. Solar energy applications are not limited to renewable energy sources [12]. DSCs are one of the main solar devices and are promising devices for developing future solar devices [13,14]. These devices have three major parts: a semiconductor/photosensitizer, an iodide/triiodide electrolyte, and a counter or Pt electrode [15,16]. The excitation of DSCs by solar energy starts with photon absorption by the utilized photosensitizer. This photosensitizer was experimentally adsorbed at the surface of the applied semiconductor (mainly TiO_2_) [17]. Thus, the role of the semiconductor is the substrate for photosensitizer-loading in addition to the electron-transfer layer from the photosensitizer to the external circuit. After that, the electrons could come back to the electrical cell by the counter electrode (mainly Pt), which could use these electrons to reduce the triiodide to iodide [18]. The oxidized dye could be restored by a reaction with iodide to form triiodide. Overall, the DSC could generate an electric current from solar energy, and it could be considered environmentally safe.

For the design of photoanodes, TiO_2_ has been reported to be the optimum semiconductor up to now because it fulfills some distinct physicochemical characteristics, including light scattering, surface area, and electrochemical stability [19]. The main disadvantage of these electrochemical devices is the charge recombination, which could be due to trapping sites because of grain boundaries [20], which could be decreased by the usage of 1D-TiO_2_ nanoparticles, including nanofibers (NFs), nanotubes, etc., [21]. 1D-TiO_2_ nanoparticles create a direct transfer for the electric current, which could increase the expected lifetime. Therefore, research work regarding 1D-TiO_2_ is an interesting field, which includes NFs photoanodes for DSCs.

Of all reported methods to fabricate NFs, electrospinning is one of the promising techniques and has different merits, such as being easy to chemically control, a simple design, and creating a pure product [22,23,24]. Furthermore, this technique could be applied to fabricate 3D networks from TiO_2_ NFs [25] or nanocomposites Zr/Ag /TiO_2_ [26], and applied for variable applications, including water treatment [27], photocatalysis [28], solar energy [29], and different multifunctional applications [30]. Additionally, many chemical treatments were reported to improve the efficiency of TiO_2_, such as surface modification [31], metal doping [32], or non-metal doping [33,34]. The N-doping strategy could enhance the optical characteristics of the utilized semiconductor in addition to improving the photo-generated electrons’ mobility via decreasing the charge recombination as well as shifting the flat-band voltage of the conduction band, which has a significant impact on both the obtained current (J_SC_) and voltage (V_OC_) [35]. According to previous studies, it is reported that N-doping enhances electron transport and broadens light absorption in the visible light area [36]. Herein, this work introduces non-metal doping or N-doping to check the performance of TiO_2_ NFs without and after N-doping as a photoanode for DSCs. The designed cells were characterized by EIS analysis to investigate the series and charge-transfer resistance, and indicate the best method to carry out the fitting of EIS data. The results indicate the importance of the existence of two-charge transfer resistances in the equivalent circuit as well as the enhancement of charge transfer after N-doping.

## 2. Materials and Methods

### 2.1. Materials and Methods

We purchased the chemicals from Aldrich, USA, including titanium (IV) isopropoxide; N,N-dimethylformamide; glacial acetic acid; poly (vinyl acetate) (PVAc); isopropyl alcohol; and chloroplatinic acid hexahydrate. We obtained FTO glass substrates from Pilkington, USA (conductive side has resistance ~8 Ω/cm). We prepared the triiodide/iodide electrolyte by a solution of 0.03 M I2, 0.60 M 1-methyl-3-butylimidazolium iodide, 0.10 M guanidinium thiocyanate, and 0.50 M 4-tert-butylpyridine, in a solvent of acetonitrile/valeronitrile (85:15).

### 2.2. Synthesis of TiO_2_ NFs and N-TiO_2_ NFs

Firstly, we prepared 14% of the used polymer (PVAc) by dissolving it in dimethylformamide (DMF) as a solvent. Then, we added glacial acetic acid drops to acidify the solution. Next, we added Ti source (titanium isopropoxyl). We stirred the obtained sol–gel at 40 °C, followed by electrospinning at potential around 12 kV and 16 cm as a distance separation between the collecting electrode and syringe. We initially dried the obtained fibers at 60 °C for 72 h. Finally, we saved the fibers for 2 h at 500 °C to eliminate the polymer contents and chemically convert the Ti substrate to its oxide. We carried out the N-doping process via hydrothermal technique in aqueous urea solution in a 200 mL Teflon container held in a stainless-steel vessel. We carried out the hydrothermal process at 180 °C for 5 h. Last, we filtered the hydrothermal product mixture and dried the obtained fibers for 1 day at 60 °C. We symbolized this as N-TiO_2_ NFs.

### 2.3. Materials Characterization and EIS Analysis

We carried out the SEM and TEM morphology of the fabricated materials by scanning electron microscopy (SEM, Hitachi S-7400, Hitachi, Japan) and transmission electron microscopy (TEM), respectively. We investigated the crystallinity by a Rigaku X-ray diffractometer (Rigaku Co., Akishima, Japan) with Cu Ka (k = 1.5405 6 Å) radiation. Furthermore, we studied the surface chemistry via X-ray photoelectron spectroscopy analysis (XPS) analysis, which was performed under these conditions: pressure, 6.5 × 10^−9^ Torr; scan step, 0.05 eV/step and resolution (pass energy); 20 eV. We fabricated the DSCs based on TiO_2_ NFs and N-TiO_2_ NFs using exactly the previously reported procedures [37]. Figure 1 shows the main components of the fabricated DSCs. We fixed the frequency range of the EIS analysis at 0.1–100 K Hz.

## 3. Results and Discussion

### 3.1. Photoanode Morphological and Chemical Characterization

The morphology of the designed photoanode material was described via SEM and TEM before and after N-doping, as shown in Figure 2. Figure 2A shows the SEM image of the electrospun fibers after calcination. The fibrous characteristics have intertwined fibers with a diameter of around 100–250 nm. The calcination step has an outstanding role in obtaining the oxide of Ti metal and, according to the observed SEM image, it does not have any negative impacts on the fibrous characteristics [38,39]. After N-doping, the SEM image was screened and described in Figure 2B, which shows the same morphological characteristics. According to previous studies, N-doping could affect the chemistry of the produced fibers to have N without any distortion in the expected morphology [17,40]. The TEM surface morphology of TiO_2_ NFs after calcination was found as nanoparticles connected to form a fibrous morphology, as shown in Figure 2C. After N-doping, the fiber was found to be smoother with no secondary particles, as seen in the TEM image of N-TiO_2_ NFs. The N-doping could play a major role in filling the pores that exist in the pristine TiO_2_ (Figure 2C) and, therefore, the fiber appeared for N-TiO_2_ NFs as smoother (Figure 2D). In short, the N-TiO_2_ NFs morphology becomes more smooth than pristine TiO_2_ NFs, as observed from the difference between the TEM images of TiO_2_ NFs (Figure 2C) and N-TiO_2_ NFs (Figure 2D). To conclude, the morphology of the fabricated TiO_2_ NFs and N-TiO_2_ NFs for the photoanode substrate in the DSCs clarifies that the designed material has a nanoscale fibrous morphology without any negative impacts after N-doping.

The XRD analyses of the synthesized TiO_2_ and N-TiO_2_ NFs materials were studied to describe the crystallinity of the fabricated NFs. Figure 3 displays the XRD analysis of the TiO_2_ and N-TiO_2_ NFs. Based on the XRD data, the peaks were detected at 25.05, 36.51, 37.23, 38.18, 47.45, 53.32, 54.61, and 62.16, which could be because of the crystal planes of anatase TiO_2_ ((101), (103), (004), (112), (200), (105), (211), and (204), respectively). The seen diffraction peaks were compared with JCPDS card No. 21-1272 [17,39], as shown in Figure 3. Similar positions for 2θ were detected for both TiO_2_ and N-TiO_2_ NFs in addition to JCPDS card No. 21-1272. The main peak at 25.05 displayed a significant line broadening, which affirms the enhancement of crystallinity after N-doping, as shown in the inset figure of Figure 3. At the main peak (2θ around 25.05), the FWHM values of TiO_2_ and N-TiO_2_ NFs are 3.5 Å and 1.96 Å, respectively, which affirm the narrowing of the N-doping peak. At the secondary peak (2θ around 47.45), the FWHM values of TiO_2_ and N-TiO_2_ NFs are 3.94 Å and 2.36 Å, respectively, which indicate the same conclusions of the crystallinity improvement after N-doping. The results indicate that the fabricated TiO_2_ and N-TiO_2_ NFs have the same crystal behaviors as JCPDS card No. 21-1272, albeit with a slight enhancement of crystallinity after hydrothermal treatment.

When investigating the surface chemistry of the NFs in addition to affirming the successful N-doping, XPS analysis was supported and is shown in Figure 4A. From XPS data, the peaks assigned to O 1s [41], N 1s [42], and Ti 2p [37] could be seen at 529.1 eV, 398.9 eV, and 457.9 eV, respectively. These peaks, as well as XRD analysis, indicate the existence of anatase TiO_2_, and provided acceptable proof of the successful N-doping by the XPS peak of N 1s at 398.9 eV [42]. The nitrogen peak was studied at the fine spectrum, as shown in Figure 4B. Two clear deconvoluted peaks could be seen at 399.07 eV and 398.48 eV, which could be assigned to different N-bonds, and were expected to be related to the substitutional N instead of O atoms in the anatase crystal of TiO_2_ NFs and N atoms inside the interstitial locations between the anatase crystals that form Ti–O–N [43]. According to Koli et al. [43], the major peak located at 399.07 eV (higher BE) could be attributed to the substitutional N atoms inside the TiO_2_ crystals, and a second peak at 398.48 eV could be due to the N atoms in the interstitial locations that form a Ti–O–N bond. Based on the XPS analysis, the atomic percentage of nitrogen is 4.37%. The major content of the designed TiO_2_ NFs is oxygen, which is due to the calcination step and forming of TiO_2_. After that, the N content is introduced after the hydrothermal step in the urea. The other small narrow peak is located at 400.78 eV, and could be due to another type of nitrogen at the depth of 10 nm, which is created by trapping N_2_ molecules inside the anatase crystals of TiO_2_ [44,45]. This peak has no impact on the discussion and conclusions regarding the larger peaks discussed above. The N species at the interstitial locations between anatase crystals could play a main role in filling the pores present in the pure TiO_2_ (Figure 2C); moreover, the fiber appeared for N-TiO_2_ NFs as smoother (Figure 2D). In short, the N-TiO_2_ NFs’ morphology becomes more smooth than pristine TiO_2_ NFs, as observed from the difference between the TEM images of TiO_2_ NFs (Figure 2C) and N-TiO_2_ NFs (Figure 2D). Additionally, the XPS analysis indicates the surface chemistry of the synthesized N-TiO_2_ NFs to be TiO_2_ and doped by nitrogen atoms.

### 3.2. Electrochemical Impedance Investigation

The DSCs based on the fabricated TiO_2_ and N-TiO_2_ NFs as a photoanode material were investigated by electrochemical impedance spectroscopy (EIS). To explain the charge transfer resistance (R_ct_) of DSCs at both photoanode and counter electrode sides in triiodide/iodide medium, our EIS investigation was conducted in terms of fitting methodology, Rct analysis, and the impacts of N-doping. To determine the series/ohmic resistance (Rs) and Rct, the fitting of the EIS data could be performed by the applied equivalent circuit. In the DSCs analysis, three simple equivalent circuits (EQ) were investigated to indicate the optimum one according to the DSC mechanism and the estimated error. These circuits were studied for DSCs and symbolized as EQ1, EQ2, and EQ3 [17,37,46,47,48]. The EIS investigation was studied at 25 °C and a frequency range of 0.1 to 100 K Hz. Figure 5 displays a comparison between the pristine Nyquist plots before and after fitting by the use of EQ1 (Figure 5A), EQ2 (Figure 5B), and EQ3 (Figure 5C) in addition to the applied circuits (Figure 5D). The electrical parameters of the applied circuits, which were obtained after fitting, are shown in Table 1. In these data, Rs is the series resistance, which is an ohmic resistance and related to the iodine electrolyte, which is between the photoanode (TiO_2_/dye) and cocountersne (Pt) [46]. In addition to this ohmic impedance part, there are other components (R_ct_ and constant phase element (CPE); R_ct1_/CPE1 and R_ct2_/CPE2) that could be assigned to the redox processes [17,49,50]. The CPE is the double-layer capacitance between the faradic and ohmic processes. The application of CPE instead of a traditional capacitor is attributed to the heterogeneity of the electrochemical device, including the counter electrode and photoanode [37,51]. The optimum equivalent circuit among the studied circuits (Figure 5A–C) is the EQ2 for many reasons. Firstly, the scientific shape of the Nyquist plot after fitting for both EQ1 has only one semi-circle instead of two if compared with the Nyquist plot without fitting; therefore, the EQ1 could be excluded. Conversely, the EQ2 and EQ3 have two semi-circles, which is similar to what was observed in the pristine data without fitting. Secondly, the fitted data of EQ2 (Figure 5B) was much closer to the data without fitting than the case of EQ3 (Figure 5C). Thirdly, the Bode plots displayed two peaks in the case of the data without fitting, which was additionally observed in the case of fitting by EQ2 or EQ3 (Figure 6A). However, EQ1 displayed only a single peak. Furthermore, the errors of the Rs were higher in the cases of EQ1 or EQ3 (4.28 and 0.41 %, respectively), and lower if EQ2 was used in the EIS fitting (0.36 %). The high value of the estimated error was additionally seen in the error values of Rct in the cases of EQ1 or EQ3 if compared with EQ2 (Figure 6B). It is noteworthy to mention that the main reason for the worse fitting when EQ1 and EQ3 were applied for the EIS fitting of DSC could be due to the complexity of the electron transfer mechanism, which depends on two electrodes: a photoanode one (TiO_2_/photosensitizer) and a counter one (mainly Pt) [8,17,46]. EQ1 might be applied when the electrochemical process has only a single charge transfer process, and EQ3 could be utilized when there are bi-dependent charge transfer processes [2]. EQ2 may be accurate when there are two separated charge transfer processes, such as DSCs. The first one is at the photoanode (semiconductor/dye electrode), and the second is between the iodine electrolyte/Pt electrode [2,17,52]. Similar analyses were carried out using N-TiO_2_ NFs instead of TiO_2_ NFs, as shown in Figure 7, and EQ2 was found as the optimum one. Therefore, N-doping has no considerable impact on the choice of the optimum equivalent circuit.

### 3.3. Effect of N-Doping on the EIS

EIS measurements, including Nyquist plots (Figure 8A) and the Bode-phase analysis (Figure 8B) of DSCs, were investigated without and after N-doping over the TiO_2_ NFs photoanode. The observed Nyquist plots have two semicircles, which could be assigned to two charge transfer resistances, as previously discussed. The first one at high-frequency values could be attributed to the charge transport at the electrolyte/Pt electrode, and the second one corresponds to resistance at the electrolyte/photoanode [17]. There are higher impedance values in the Nyquist characteristics of the N-TiO_2_ photoanode if compared with the TiO_2_ photoanode without N-doping. The R_ct1_ increased from 5.635 ohm.cm^2^ to 1.054 ohm.cm2, which could be due to the N-doping. Figure 8B describes the Bode phase as well as total impedance based on the TiO_2_ photoanode without N-doping and the N-TiO_2_ photoanode after N-doping. As seen in Figure 8B, electron lifetimes (τ_n_) can be estimated by the following reported equation [15]: τ_n_ = 1/(2πf_peak_)(1)

The f_max_ values for the TiO_2_ and N-TiO_2_ photoanodes were found at 12.589 Hz and 15.849 Hz, respectively, and the calculated electron lifetime was slightly decreased from 12.64 ms to 10.04 ms after N-doping, which had a slightly negative impact over the electron lifetime, which is in accordance with previous studies and could be attributed to the electron/hole (charge) recombination [37]. However, the decrease of charge transfer resistance has a higher impact if compared with the slight decrease of electron lifetime. Furthermore, the total impedance of the DSC is additionally described in Figure 8B without and after N-doping. There is a clear trend toward decreasing impedance values after N-doping, which confirms the improved general electrical conductivity of the N- TiO_2_ photoanode substrate if compared with the TiO_2_ photoanode substrate without N-doping. Lastly, the structural changes could be correlated with photoanode performance. After N-doping, the TEM and XRD analyses indicated the formation of a smoother fiber with higher crystallinity, which filled the pores by N-doping, which could play a positive role in the dye-loading of the TiO_2_ substrate; therefore, more dye molecules could be adsorbed at the N-TiO_2_ surface compared with pure TiO_2_. The enhancement of dye-loading after N-doping was reported by previous studies, including Refs. [35,53,54]. In short, N-doping could improve dye-loading, in addition to the electron transfer of the photoanode characteristics of DSCs.

## 4. Conclusions

In short, electrochemical impedance (EIS) was studied for DSCs using TiO_2_ and N-TiO_2_ NFs as a photoanode substrate. The equivalent circuit fitting analysis was carried out by different reported equivalent circuits, and we found that the optimum circuit was an equivalent circuit with two separated Rct. The two-charge transfer process could be assigned to the one at the photoanode (semiconductor/dye electrode) and the second between the iodine electrolyte/Pt electrode. The Rct1 increased from 5.635 ohm.cm^2^ to 1.054 ohm.cm^2^, which could be due to the N-doping. To summarize, this work introduces a simple way toward equivalent circuit fitting and confirms the impact of N-doping on the EIS of DSCs.

## Figures and Tables

**Figure 1 materials-15-06175-f001:**
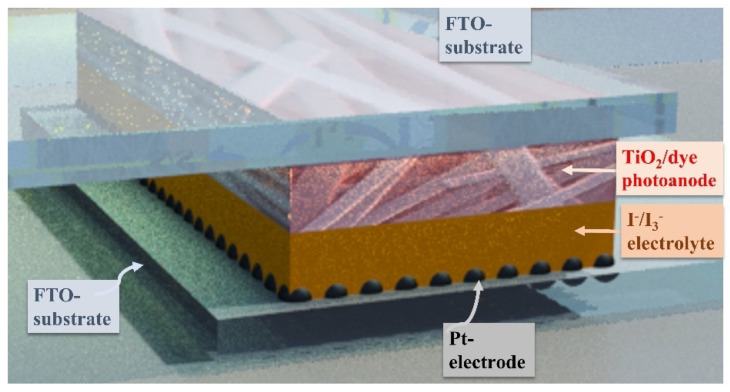
Schematic diagram of the fabricated solar cell based on the synthesized nanofibers.

**Figure 2 materials-15-06175-f002:**
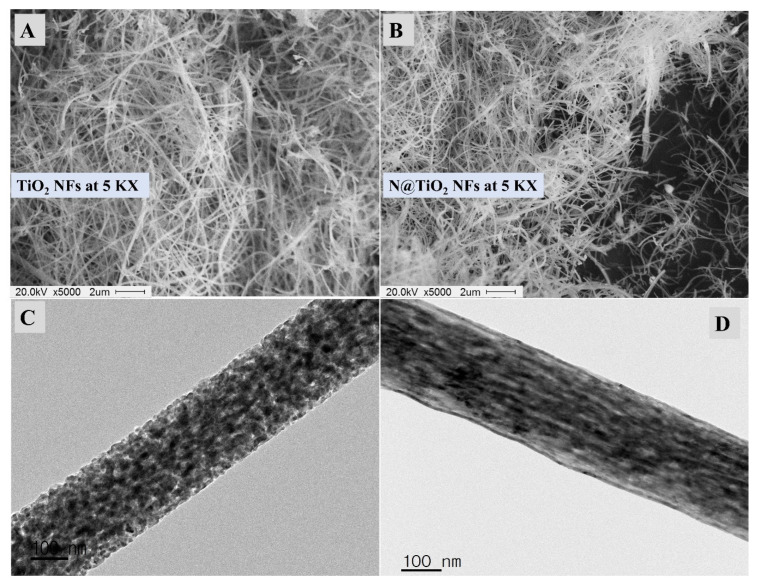
(**A**) SEM image of the prepared Titania; (**B**) SEM image of N-Titania; (**C**) TEM image of the prepared Titania; (**D**) TEM image of the prepared N-Titania.

**Figure 3 materials-15-06175-f003:**
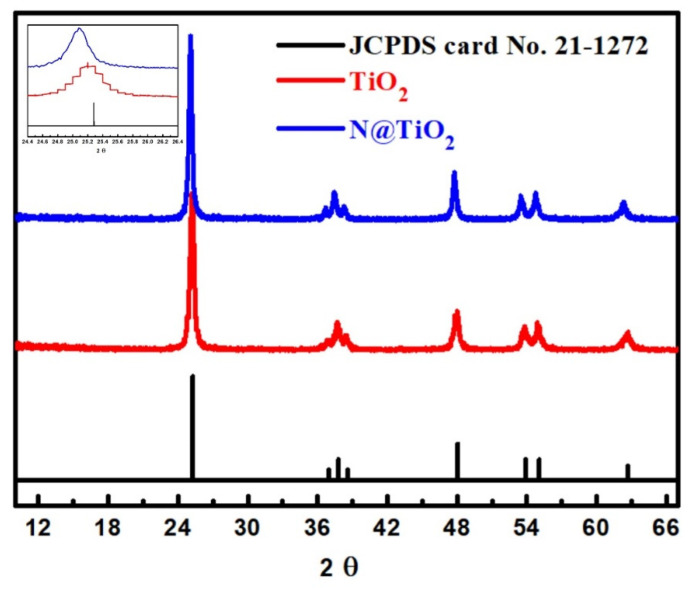
XRD analysis of Titania and N-Titania in addition to the locations of JCPDS card No. 21-1272. The inset figure is an overlay of the 25.05 peaks for TiO_2_ and N-TiO_2_ to better visualize the line broadening.

**Figure 4 materials-15-06175-f004:**
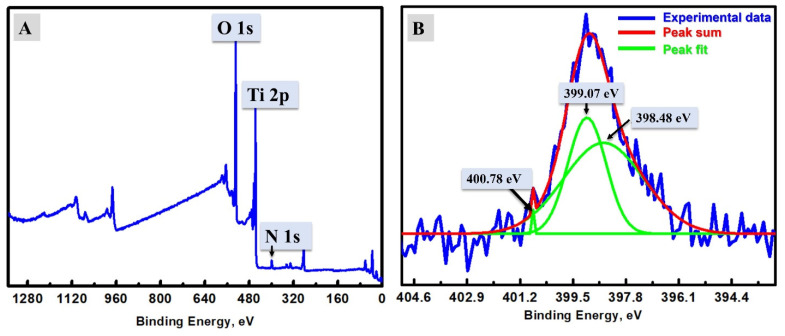
(**A**) XPS analysis of N-Titania; (**B**) XPS fine spectrum of the N-Titania in the region of nitrogen.

**Figure 5 materials-15-06175-f005:**
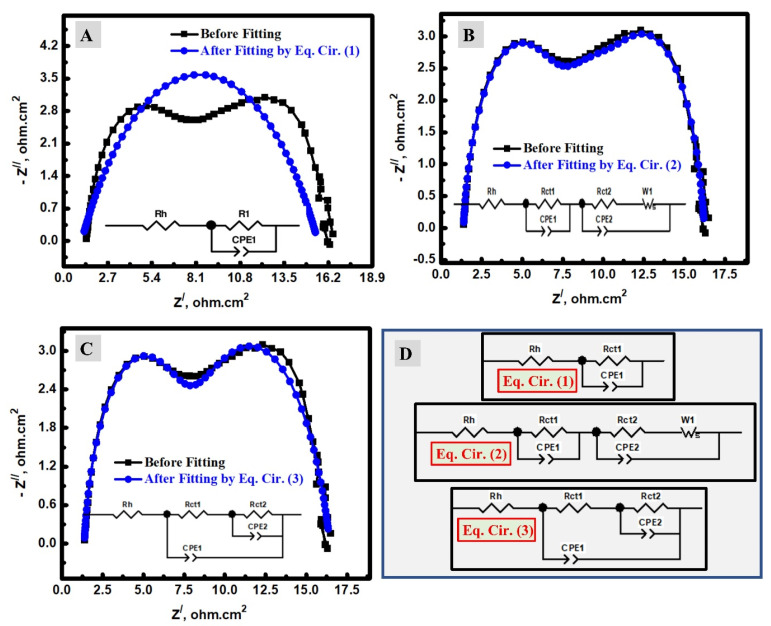
EIS analyses of DSCs using the prepared TiO_2_ NFs, including experimental data and their fitting by (**A**) EQ1; (**B**) EQ2; (**C**) EQ3; (**D**) utilized equivalent circuits.

**Figure 6 materials-15-06175-f006:**
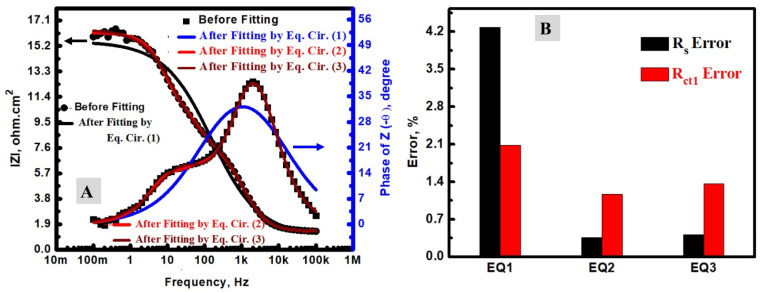
(**A**) Bode phase and total impedance data; (**B**) the estimated R_s_ and R_ct_ errors (%).

**Figure 7 materials-15-06175-f007:**
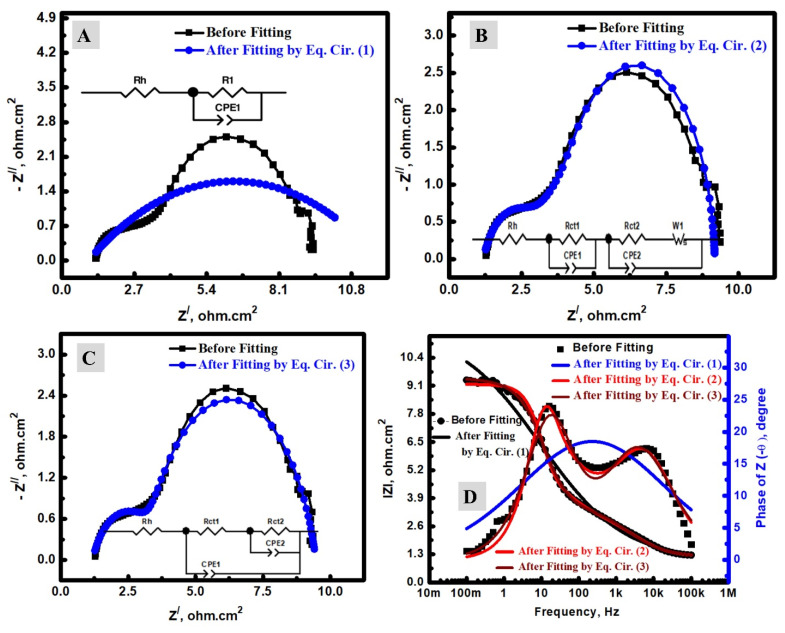
EIS analyses of DSCs using the prepared N-TiO_2_ NFs, including experimental data and their fitting by (**A**) EQ1; (**B**) EQ2; (**C**) EQ3; (**D**) total impedance and bode phase data before and after equivalent circuit fitting.

**Figure 8 materials-15-06175-f008:**
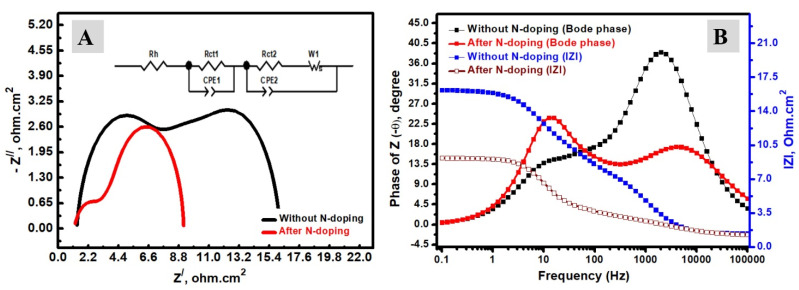
(**A**) EIS analyses of DSCs without and after N-doping of TiO_2_ photoanode; (**B**) Bode phase and total impedance data N-doping of TiO_2_ photoanode.

**Table 1 materials-15-06175-t001:** Electrochemical impedance parameters found after EIS fitting using different equivalent circuits (Figure 5D).

No.	EQ	R_s_ ohm.cm^2^	R_ct1_ ohm.cm^2^	R_ct2_ ohm.cm^2^	CPE1-T µF	CPE1-P, F	CPE2-T(+), µF	CPE2-P(+), F
1	EQ1	1.09	14.46	-	374.2	0.586	-	-
Error, %		4.2823	2.0859	-	12.792	2.638	-	-
2	EQ2	1.3705	5.635	5.525	21.423	0.89	524.73	0.794
Error, %		0.3614	1.1652	1.39	5.218	0.95	38.25	8.48
3	EQ3	1.3688	6.72	8.35	21.53	0.881	1228.3	0.72
Error, %		0.4176	1.3679	1.50	4.98	0.57	4.27	1.599

## Data Availability

The raw/processed data generated in this work are available upon request from the corresponding author.

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
