# Peer review of "Electrochemical Impedance Investigation of Dye-Sensitized Solar Cells Based on Electrospun TiO2 Nanofibers Photoanodes"

_materials, 2022, doi:10.3390/ma15176175_

Round 1

Reviewer 1 Report

This study evaluates the electrochemical impedance for different photoanode substrates. Overall the results are satisfactory and are supported by enough data. I recommend publication after addressing my following concern.

Figures are poorly drawn, especially Figures 5, 6,7 8 are not correctly aligned. 

Author Response

AUTHOR’S RESPONSE TO REVIEWER # 1 COMMENTS:

< materials-1855674>

< Electrochemical impedance investigation of dye-sensitized solar cells based on electrospun TiO2 nanofibers photoanodes >

Dear Editor of “Materials”

Firstly, we would like to thank you for giving us a chance to resubmit the work after revision, and also thank the reviewers for giving us constructive suggestions which would help us to improve the quality of the paper. Here we submit a revised version of our manuscript, which has been modified according to the editor and reviewers’ suggestions. We mark the changes in yellow color in the new version manuscript. The detailed corrections are listed below point by point:  

RESPONSE TO REVIEWER # 1 COMMENTS: 

This study evaluates the electrochemical impedance for different photoanode substrates. Overall, the results are satisfactory and are supported by enough data. I recommend publication after addressing my following concern.

We would like to thank the reviewer for his great efforts and for giving useful criticism to the article.

Figures are poorly drawn, especially Figures 5, 6,7 8 are not correctly aligned.

Author reply: Thanks for the reviewer’s comment. Figures 5, 6, 7, and 8 were redesigned as suggested by the reviewer.

Thank you for your efforts and comments, we really find the comment valid and useful.

I hope that all modifications have taken into account the suggestions of the editor and reviewers. The manuscript has been resubmitted to your journal. We look forward to your positive response.   

Sincerely,  

Prof. Hany M. Abd El-Lateef

Reviewer 2 Report

The manuscript by Abd El-Lateef et al. evaluates the electrochemical impedance behaviour of dye-sensitized solar cells based on electrospun TiO2 nanofibers photoanodes. The manuscript is submitted to the special issue Development of Nano-Materials for Catalytic and Biomedical Applications, and fits well to the catalytic applications.

The following aspects can be improved, as described below:

1. Authors can indicate in the introduction why non-metal doping or N- doping was done - this would provide readers with better insight.

2.  Figure 2 C,D: having a look at the TEM images of the NFs, the TiO2 NFs seem to have smaller grains, while these are not as sharp or larger for N-TiO2. Is this a matter of the N treatment or a result from focus in TEM?

3.  Figure 3 XRD - authors can include an inset with a overlay of the 25.05 peak for  TiO2 and N-TiO2, to better visualize the line broadening and support the discussion at ln 135-139.

4.  XPS - Figure 4 & discussion: a) survey of Fig 4a - there is a doublet peak at around 320-340eV, and one peak in between 240-320eV which are not assigned. Please include these peaks too. b) peak fitting in Fig 4b - what is the smaller peak (very very narrow FWHM) at 400.78eV. In addition, authors could include the N-peak for TiO2, which should be very small and just adsorbed N.

5. Other minor things: ln 97 - title of section 2.3; some arreas minor English corrections are needed

Author Response

AUTHOR’S RESPONSE TO REVIEWER # 2 COMMENTS:

< materials-1855674>

< Electrochemical impedance investigation of dye-sensitized solar cells based on electrospun TiO2 nanofibers photoanodes >

Dear Editor of “Materials”

Firstly, we would like to thank you for giving us a chance to resubmit the work after revision, and also thank the reviewers for giving us constructive suggestions which would help us to improve the quality of the paper. Here we submit a revised version of our manuscript, which has been modified according to the editor and reviewers’ suggestions. We mark the changes in yellow color in the new version manuscript. The detailed corrections are listed below point by point:  

RESPONSE TO REVIEWER # 2 COMMENTS: 

The manuscript by Abd El-Lateef et al. evaluates the electrochemical impedance behavior of dye-sensitized solar cells based on electrospun TiO2 nanofibers photoanodes. The manuscript is submitted to the special issue Development of Nano-Materials for Catalytic and Biomedical Applications and fits well to the catalytic applications.

The following aspects can be improved, as described below:

We would like to thank the reviewer for his great efforts and for giving useful criticism to the article.

  1. Authors can indicate in the introduction why non-metal doping or N- doping was done - this

would provide readers with better insight.

Author reply: Thanks for the reviewer’s comment. The introduction was supported by the following according to the suggestion from the reviewer “N-doping strategy could enhance the optical characteristics of the utilized semiconductor in addition to improvement of the photo-generated electrons mobility via decreasing the charge recombination as well as shifting the flat-band voltage of conduction band which has a significant impact on both the obtained current (JSC) and voltage (VOC) [1]. According to previous studies, it is reported that N-doping enhances electron transport and broadens light absorption in the visible light area [2]. Herein, this work introduces non-metal doping or N-doping to check the performance of TiO2 NFs without and after N-doping as photoanode for DSCs. The designed cells were characterized by EIS analysis to investigate the series, charge-transfer resistance, and indicate what is the best method to carry out the fitting of EIS data”.

  1. Figure 2 C,D: having a look at the TEM images of the NFs, the TiO2 NFs seem to have smaller grains, while these are not as sharp or larger for N-TiO2. Is this a matter of the N treatment or a result from focus in TEM?

Author reply: Thanks for the reviewer’s comment. The morphology of the designed photoanode material was described via SEM and TEM before and after N-doping as shown in Fig. 2. Fig. 2C shows the TEM image of the electrospun fibers after calcination. The fibrous characteristics have intertwined fibers with a diameter of around 250 nm. After N-doping, the TEM image was screened and described in Fig. 2D which shows the same fibrous morphological characteristics except a more smooth character was seen after hydrothermal or N-doping. According to previous studies, N-doping could affect the chemistry of the produced fibers to have N without any distortion in the expected morphology [3, 4]. The smooth surface with no secondary particles was seen in the TEM image of N-TiO2 NFs as displayed in 2D. After N-doping, the fiber was found as a smoother fiber with no secondary particles as seen in the TEM image of N-TiO2 NFs. The N-doping could play a major role to fill the pores existent in the pristine TiO2 (Fig. 2C) and so, the fiber appeared for N-TiO2 NFs as smoother (Fig. 2D). The morphology of the fabricated TiO2 NFs and N-TiO2 NFs for photoanode substrate in DSCs clarifies that the designed material has nanoscale fibrous morphology without any negative impact after N-doping. In short, there is no change in fibrous morphology after N-doping according to SEM or TEM images.

According to XPS, there are three types of nitrogen were existed in the prepared N-TiO2 NFs. Two clear deconvoluted peaks could be seen at 399.07 eV, and 398.48 eV which could be assigned to different N-bonds and expected to be related to the substitutional N instead of O-atoms in the anatase crystal of TiO2 NFs, and N atoms inside the interstitial locations between anatase crystals to form Ti–O–N [5]. The other small narrow peak located at 400.78 eV and could be due to another type of nitrogen at the depth of 10 nm which is the trapping N2-molecules inside the anatase crystals of TiO2. The N-species at the interstitial locations between anatase crystals and N2-molecules inside the TiO2 crystals could play the main role to fill the pores present in the pure TiO2 (Fig. 2C) and so, the fiber appeared for N-TiO2 NFs as smoother (Fig. 2D). In short, the N-TiO2 NFs morphology becomes more smooth than pristine TiO2 NFs as observed from the difference between TEM image of TiO2 NFs (Fig. 2C) and N-TiO2 NFs (Fig. 2D). This discussion was added in the revised version of the manuscript.

  1. Figure 3 XRD - authors can include an inset with an overlay of the 25.05 peak for TiO2 and N-TiO2, to better visualize the line broadening and support the discussion at ln 135-139.

Author reply: Thanks for the reviewer’s comment. The main peak at 25.05 displayed a significant line broadening which affirms the enhancement of crystallinity after N-doping as shown in the inset figure of Fig. 3. The inset figure of Figure 3 was supported in the revised version of the manuscript according to the reviewer’s advice.

  1. XPS - Figure 4 & discussion: a) survey of Fig 4a - there is a doublet peak at around 320-340eV, and one peak in between 240-320eV which are not assigned. Please include these peaks too. b) peak fitting in Fig 4b - what is the smaller peak (very narrow FWHM) at 400.78eV. In addition, authors could include the N-peak for TiO2, which should be very small and just adsorbed N.

Author reply: Thanks for the reviewer’s comment. To investigate the surface chemistry of the NFs in addition to affirming the successful N-doping, XPS analysis was supported and shown in Fig. 4A. From XPS data, the peaks assigned to O 1s [6], N 1s [7], and Ti 2p [8] could be seen at 529.1 eV, 398.9 eV, and 457.9 eV, respectively. There are some other peaks such as double peaks at around 320-340eV which were exactly at 336 eV and 346 eV in addition to one peak between 240-320eV (exactly at 284 eV). All these peaks and other small peaks in the XPS spectrum survey were existence because of surface contamination of the investigated material during experimental XPS measurements. The peak at 284 eV could be assigned to carbon contamination which is present on the surface as CO2. XPS spectra peaks are obtained by irradiating a solid surface with a beam of X-rays and measuring the kinetic energy of electrons that are emitted from the top 1-10 nm of the material. Therefore, gas contamination at this depth could occur. To remove this contamination, a plasma cleaning strategy should be applied before XPS analysis which isn’t available for us. This work introduces XPS analysis to confirm the successful doping of N after hydrothermal treatment. Thus, the discussion of XPS focus on this point.

The nitrogen peak was deeply studied at the fine spectrum as shown in Fig. 4B. Two clear deconvoluted peaks could be seen at 399.07 eV, and 398.48 eV which could be assigned to different N-bonds and expected to be related to the substitutional N instead of O-atoms in the anatase crystal of TiO2 NFs, and N atoms inside the interstitial locations between anatase crystals to form Ti–O–N [5]. The other small narrow peak is small and has no impact on the discussion and conclusions regarding the larger peaks discussed above. The spectra could have been fitted without this narrow peak, but the overall quality of the fit and the consistency between the experimental peak and peak summation should be exactly the same as much as we can. For this small peak, it could be due to another type of nitrogen at the depth of 10 nm which is the trapping N2-molecules inside the anatase crystals of TiO2.

  1. Other minor things: ln 97 - title of section 2.3; some arreas minor English corrections are

Needed

Author reply: Thanks for the reviewer’s comment. The authors corrected the error and revised all manuscript parts to be free from any editing or language errors.

Thank you for your efforts and comments, we really find the comment valid and useful.

I hope that all modifications have taken into account the suggestions of the editor and reviewers. The manuscript has been resubmitted to your journal. We look forward to your positive response.   

Sincerely,  

Prof. Hany M. Abd El-Lateef

REFERENCES

  1. Rajaramanan, T.; Kumara, G. R. A.; Velauthapillai, D.; Ravirajan, P.; Senthilnanthanan, M., Ni/N co-doped P25 TiO2 photoelectrodes for efficient Dye-Sensitized Solar Cells. Materials Science in Semiconductor Processing 2021, 135, 106062.
  2. Guo, W.; Shen, Y.; Boschloo, G.; Hagfeldt, A.; Ma, T., Influence of nitrogen dopants on N-doped TiO2 electrodes and their applications in dye-sensitized solar cells. Electrochimica Acta 2011, 56, (12), 4611-4617.
  3. Mohamed, I. M. A.; Dao, V.-D.; Yasin, A. S.; Barakat, N. A. M.; Choi, H.-S., Design of an efficient photoanode for dye-sensitized solar cells using electrospun one-dimensional GO/N-doped nanocomposite SnO2/TiO2. Applied Surface Science 2017, 400, 355-364.
  4. Yao, S.; He, Y.; Wang, Y.; Bi, M.; Liang, Y.; Majeed, A.; Yang, Z.; Shen, X., Porous N-doped carbon nanofibers assembled with nickel ferrite nanoparticles as efficient chemical anchors and polysulfide conversion catalyst for lithium-sulfur batteries. Journal of Colloid and Interface Science 2021, 601, 209-219.
  5. Koli, V. B.; Mavengere, S.; Kim, J.-S., An efficient one-pot N doped TiO2-SiO2 synthesis and its application for photocatalytic concrete. Applied Surface Science 2019, 491, 60-66.
  6. Zhang, J.; Raza, S.; Wang, P.; Wen, H.; Zhu, Z.; Huang, W.; Mohamed, I. M. A.; Liu, C., Polymer brush-grafted ZnO-modified cotton for efficient oil/water separation with abrasion/acid/alkali resistance and temperature “switch” property. Journal of Colloid and Interface Science 2020, 580, 822-833.
  7. Iqbal, W.; Yang, B.; Zhao, X.; Rauf, M.; Mohamed, I. M. A.; Zhang, J.; Mao, Y., Facile one-pot synthesis of mesoporous g-C3N4 nanosheets with simultaneous iodine doping and N-vacancies for efficient visible-light-driven H2 evolution performance. Catalysis Science & Technology 2020, 10, (2), 549-559.
  8. Mohamed, I. M. A.; Dao, V.-D.; Yasin, A. S.; Mousa, H. M.; Mohamed, H. O.; Choi, H.-S.; Hassan, M. K.; Barakat, N. A. M., Nitrogen-doped&SnO2-incoportaed TiO2 nanofibers as novel and effective photoanode for enhanced efficiency dye-sensitized solar cells. Chemical Engineering Journal 2016, 304, 48-60.

Reviewer 3 Report

This paper discusses about the electrochemical impedance studies of DSC based on TiO2 NF photoanodes. However, the following key points must be answered before accepting the manuscript for publication.

1.     Please define and abbreviate the term DMF mentioned in page no 2 of the manuscript.

2.     Units such as °C should be made uniform throughout the manuscript and rewrite hours as h to make it unique everywhere.

3.     There are few grammatical errors in the manuscript. It must be improved. For e.g in page 2, line 94, the authors stated that “The hydrothermal vessel was saved for 5 hours at 180 °C”.

4.     To compare the morphological changes in SEM before and after N incorporation, the size and resolution provided in the samples should be same.

5.     In figure 2 c, it seems there are some pores present in the TiO2 nanofibers whereas after N doping the samples appeared to be smooth. Could the authors explain the difference or support to explain this behavior in the morphological change?

6.     In the introduction, these articles can be included to extend the scope of this work to a wider audience.

(A)https://doi.org/10.1016/j.electacta.2017.02.154,

(B) https://doi.org/10.1021/acsami.0c07075

7.     In page no 3 line 135, the authors claim that line broadening leads to enhancement in crystallinity after doping. Could you explain what does the authors define here? Because from the figure it seems FWHM reduces.

8.     Why does author define FWHM as FWHL? Please state the importance of defining as FWHL rather than FWHM.

9.     The authors are requested to label all the peaks in the survey spectra of XPS as shown in Fig 4 a. In fig 4 b, the inset terms defined are not clear please rewrite it.

10.  What kind of chemical bond is existing around 400.78 eV in N 1s spectra? The authors must define it.

11.  In page 6 line 195, it should be Figure 5 not Figure 4.

12.  The authors need to correlate the structural changes with the changes in photoanode performance of DSC after N doping.

13.  The authors are requested to comment and mention the difference and importance with the work published earlier.

https://doi.org/10.1016/j.cej.2016.06.061

14.  Reference style should be made uniform.

Author Response

AUTHOR’S RESPONSE TO REVIEWER # 3 COMMENTS:

< materials-1855674>

< Electrochemical impedance investigation of dye-sensitized so-lar cells based on electrospun TiO2 nanofibers photoanodes >

Dear Editor of “Materials”

Firstly, we would like to thank you for giving us a chance to resubmit the work after revision, and also thank the reviewers for giving us constructive suggestions which would help us to improve the quality of the paper. Here we submit a revised version of our manuscript, which has been modified according to the editor and reviewers’ suggestions. We mark the changes in yellow color in the new version manuscript. The detailed corrections are listed below point by point:  

RESPONSE TO REVIEWER # 3 COMMENTS: 

This paper discusses the electrochemical impedance studies of DSC based on TiO2 NF photoanodes. However, the following key points must be answered before accepting the manuscript for publication.

We would like to thank the reviewer for his great efforts and for giving an important criticism of the article.

  1. Please define and abbreviate the term DMF mentioned in page no 2 of the manuscript.

Author reply: Thanks for the reviewer’s comment. The statement was revised, and it became “Firstly, 14% of the used polymer (PVAc) was prepared by dissolving it in dimethylformamide (DMF) as a solvent.”. So, the DMF was defined as dimethylformamide.

  1. Units such as °C should be made uniform throughout the manuscript and rewrite hours as h to

make it unique everywhere.

Author reply: Thanks for the reviewer’s comment. All units were revised to be uniform, and hours become h as suggested by the reviewer.

  1. There are few grammatical errors in the manuscript. It must be improved. For e.g in page 2, line 94, the authors stated that “The hydrothermal vessel was saved for 5 hours at 180 °C”.

Author reply: Thanks for the reviewer’s comment. All manuscript parts were revised to be free from any editing or language errors. The error on line 94 was checked and corrected. Moreover, based on the reviewer’s suggestion; we have carefully revised the manuscript by use of “premium Grammarly (https://app.grammarly.com/ddocs/427154829)”. The revised details can be found in the revised version

  1. To compare the morphological changes in SEM before and after N incorporation, the size and resolution provided in the samples should be same.

Author reply: Thanks for the reviewer’s comment. The SEM image at 5 KX for TiO2 NFs was inserted instead of the 1 KX image in the revised version of the manuscript. So, the revised version has SEM images at 5 KX for both TiO2 NFs and N-TiO2 NFs. After N-doping, the SEM image has the same fibrous morphological characteristics.

  1. In figure 2 c, it seems there are some pores present in the TiO2 nanofibers whereas after N doping the samples appeared to be smooth. Could the authors explain the difference or support to explain this behavior in the morphological change?

Author reply: Thanks for the reviewer’s comment. The N-doping was carried out by hydrothermal treatment using urea as a source of nitrogen. According to XPS, there are three types of nitrogen were exist in the prepared N-TiO2 NFs. Two clear deconvoluted peaks could be seen at 399.07 eV, and 398.48 eV which could be assigned to different N-bonds and expected to be related to the substitutional N instead of O-atoms in the anatase crystal of TiO2 NFs, and N atoms inside the interstitial locations between anatase crystals to form Ti–O–N [1]. The other small narrow peak located at 400.78 eV and could be due to another type of nitrogen at the depth of 10 nm which is the trapping N2-molecules inside the anatase crystals of TiO2. The N-species at the interstitial locations between anatase crystals and N2-molecules inside the TiO2 crystals could play the main role to fill the pores present in the pure TiO2 (Fig. 2C) and so, the fiber appeared for N-TiO2 NFs as smoother (Fig. 2D). In short, the N-TiO2 NFs morphology becomes smoother than pristine TiO2 NFs as observed from the difference between TEM image of TiO2 NFs (Fig. 2C) and N-TiO2 NFs (Fig. 2D). This discussion was added in the revised version of the manuscript.

  1. In the introduction, these articles can be included to extend the scope of this work to a wider audience. (A)https://doi.org/10.1016/j.electacta.2017.02.154, (B) https://doi.org/10.1021/acsami.0c07075

Author reply: Thanks for the reviewer’s comment. The introduction part was modified and supported by the suggested literature from the reviewer.

  1. In page no 3 line 135, the authors claim that line broadening leads to enhancement in crystallinity after doping. Could you explain what does the authors define here? Because from the figure it seems FWHM reduces.

Author reply: Thanks for the reviewer’s comment. The inset figure of Figure 3 was supported in the revised version of the manuscript to clarify the difference in FWHM. At the main peak (2θ around 25.05), the FWHM values of TiO2 and N-TiO2 NFs are 3.5 Å, and 1.96 Å respectively which affirm the narrower of the N-doping peak. At the secondary peak (2θ around 47.45), the FWHM values of TiO2 and N-TiO2 NFs are 3.94 Å, and 2.36 Å respectively which indicates the same conclusion of the crystallinity improvement after N-doping.

  1. Why does author define FWHM as FWHL? Please state the importance of defining as FWHL

rather than FWHM.

Author reply: Thanks for the reviewer’s comment and for helping us to improve the manuscript via decreasing the editing errors in the manuscript. In the revised version of the manuscript, the incorrect “FWHL” was changed to the corrected one “FWHM”.

  1. The authors are requested to label all the peaks in the survey spectra of XPS as shown in Fig 4 a. In fig 4 b, the inset terms defined are not clear please rewrite it.

Author reply: Thanks for the reviewer’s comment. As suggested by the reviewer, the inset terms of Fig. 4B became the same character of labels in Fig. 4A in the revised version of the manuscript.

  1. What kind of chemical bond is existing around 400.78 eV in N 1s spectra? The authors must define it.

Author reply: Thanks for the reviewer’s comment. The nitrogen peak was deeply studied at the fine spectrum as shown in Fig. 4B. Two clear deconvoluted peaks could be seen at 399.07 eV, and 398.48 eV which could be assigned to different N-bonds and expected to be related to the substitutional N instead of O-atoms in the anatase crystal of TiO2 NFs, and N atoms inside the interstitial locations between anatase crystals to form Ti–O–N [1]. The other small narrow peak is small and has no impact on the discussion and conclusions regarding the larger peaks discussed above. The spectra could have been fitted without this narrow peak, but the overall quality of the fit and the consistency between the experimental peak and peak summation should be exactly the same as much as we can. For this small peak, it could be due to another type of nitrogen at the depth of 10 nm which is the trapping N2-molecules inside the anatase crystals of TiO2.

  1. In page 6 line 195, it should be Figure 5 not Figure 4.

Author reply: Deep thanks for the reviewer’s comment and for helping us to find the error. The figure number was corrected in the revised version of the manuscript.

  1. The authors need to correlate the structural changes with the changes in photoanode performance of DSC after N doping.

Author reply: Thanks for the reviewer’s comment. For structural changes after N-doping, the TEM and XRD analysis indicates the formation of smoother fiber with higher crystallinity and filling the pores by N-doping which could play a positive role in the dye-loading of TiO2 substrate and so, more dye molecules could be adsorbed at the N-TiO2 surface better than pure TiO2. The enhancement of dye-loading after N-doping was reported by different previous studies including (Materials Science in Semiconductor Processing 135 (2021): 106062), (Journal of Photochemistry and Photobiology A: Chemistry 349 (2017): 63-72), (Solar energy materials and solar cells 117 (2013): 624-631) [2-4]. This discussion was added to the revised version of the manuscript.

  1. The authors are requested to comment and mention the difference and importance with the work published earlier. https://doi.org/10.1016/j.cej.2016.06.061

Author reply: Thanks for the reviewer’s comment. This work investigates the electrochemical impedance analysis based on synthesized TiO2 and N-TiO2 NFs photoanode. This work focuses on EIS and equivalent circuit fitting by different equivalent circuits and studies the impact of N-doping which was confirmed via XPS analysis. N-doping has an outstanding role in the enhancement of charge transfer which could be due to the strong interaction between N-atoms as active sites and the used photosensitizer. The previous manuscript (https://doi.org/10.1016/j.cej.2016.06.061) study nitrogen doped&SnO2 co-incorporated TiO2 to be used as photoanode material. This previous study investigates the influence of SnO2 besides nitrogen doping. The similarity between this work and the previous one focuses on only N-doping which was reported in many studies not only in these two studies such as (Journal of Alloys and Compounds 870 (2021): 159527), (Journal of Alloys and Compounds 659 (2016): 15-22), (The Journal of Physical Chemistry C 119.29 (2015): 16552-16559), (Solar energy materials and solar cells 94.10 (2010): 1582-1590), (Electrochimica Acta 93 (2013): 202-206.), (Application of N-doped TiO₂/Bi₂O₃/rGO nanocomposites as a photoanode material in dye-sensitized solar cells. Diss. 2021), (Electrochimica Acta 56(12), 2011, Pages 4611-4617), (Solar Energy Materials and Solar Cells 117, 2013, Pages 624-631), (Nano Energy 2.5 (2013): 545-552), (RSC advances 4.20 (2014): 9946-9952), (Journal of Photochemistry and Photobiology A: Chemistry 219.2-3 (2011): 180-187), (Materials Chemistry and Physics 124.1 (2010): 422-426), (Rsc Advances 4.89 (2014): 48236-48244), (Journal of Power Sources 306 (2016): 764-771), (Int. J. Electrochem. Sci 8.6 (2013): 7984-7990), (Applied Surface Science 400 (2017): 355-364), (International journal of energy research 38.7 (2014): 908-917), (Electrochimica Acta 115 (2014): 493-498), (Applied Surface Science 399 (2017): 515-522), (RSC Advances 4.33 (2014): 16992-16998), (Applied Surface Science 269 (2013): 55-59), (Journal of Electronic Materials 47.10 (2018): 6241-6250), (Journal of Power Sources 300 (2015): 254-260) [5-28]. The N-doping at the semiconductor material towards photoanodes of dye-sensitized solar cells could be considered an interesting and promising research point so, many articles were published before and we think many articles will be published in the future in this research point.

  1. Reference style should be made uniform.

 Author reply: Thanks for the reviewer’s comment. The reference style becomes uniform in the revised version of the manuscript.

Thank you for your efforts and comments, we really find the comments valid and useful.

I hope that all modifications have taken into account the suggestions of the editor and reviewers. The manuscript has been resubmitted to your journal. We look forward to your positive response.   

Sincerely,  

Prof. Hany M. Abd El-Lateef

REFERENCES

  1. Koli, V. B.; Mavengere, S.; Kim, J.-S., An efficient one-pot N doped TiO2-SiO2 synthesis and its application for photocatalytic concrete. Applied Surface Science 2019, 491, 60-66.
  2. Dissanayake, M. A. K. L.; Kumari, J. M. K. W.; Senadeera, G. K. R.; Thotawatthage, C. A.; Mellander, B. E.; Albinsson, I., A novel multilayered photoelectrode with nitrogen doped TiO2 for efficiency enhancement in dye sensitized solar cells. Journal of Photochemistry and Photobiology A: Chemistry 2017, 349, 63-72.
  3. Rajaramanan, T.; Kumara, G. R. A.; Velauthapillai, D.; Ravirajan, P.; Senthilnanthanan, M., Ni/N co-doped P25 TiO2 photoelectrodes for efficient Dye-Sensitized Solar Cells. Materials Science in Semiconductor Processing 2021, 135, 106062.
  4. Melhem, H.; Simon, P.; Wang, J.; Di Bin, C.; Ratier, B.; Leconte, Y.; Herlin-Boime, N.; Makowska-Janusik, M.; Kassiba, A.; Boucle, J., Direct photocurrent generation from nitrogen doped TiO2 electrodes in solid-state dye-sensitized solar cells: Towards optically-active metal oxides for photovoltaic applications. Solar energy materials and solar cells 2013, 117, 624-631.
  5. Gao, Y.; Feng, Y.; Zhang, B.; Zhang, F.; Peng, X.; Liu, L.; Meng, S., Double-N doping: a new discovery about N-doped TiO 2 applied in dye-sensitized solar cells. RSC Advances 2014, 4, (33), 16992-16998.
  6. Mohamed, I. M.; Dao, V.-D.; Yasin, A. S.; Barakat, N. A.; Choi, H.-S., Design of an efficient photoanode for dye-sensitized solar cells using electrospun one-dimensional GO/N-doped nanocomposite SnO2/TiO2. Applied Surface Science 2017, 400, 355-364.
  7. Kang, S. H.; Kim, H. S.; Kim, J.-Y.; Sung, Y.-E., Enhanced photocurrent of nitrogen-doped TiO2 film for dye-sensitized solar cells. Materials Chemistry and Physics 2010, 124, (1), 422-426.
  8. Guo, W.; Wu, L.; Chen, Z.; Boschloo, G.; Hagfeldt, A.; Ma, T., Highly efficient dye-sensitized solar cells based on nitrogen-doped titania with excellent stability. Journal of Photochemistry and Photobiology A: Chemistry 2011, 219, (2-3), 180-187.
  9. Park, J.-Y.; Lee, K.-H.; Kim, B.-S.; Kim, C.-S.; Lee, S.-E.; Okuyama, K.; Jang, H.-D.; Kim, T.-O., Enhancement of dye-sensitized solar cells using Zr/N-doped TiO 2 composites as photoelectrodes. RSC advances 2014, 4, (20), 9946-9952.
  10. Zhang, Y.-Q.; Ma, D.-K.; Zhang, Y.-G.; Chen, W.; Huang, S.-M., N-doped carbon quantum dots for TiO2-based photocatalysts and dye-sensitized solar cells. Nano Energy 2013, 2, (5), 545-552.
  11. Melhem, H.; Simon, P.; Wang, J.; Di Bin, C.; Ratier, B.; Leconte, Y.; Herlin-Boime, N.; Makowska-Janusik, M.; Kassiba, A.; Bouclé, J., Direct photocurrent generation from nitrogen doped TiO2 electrodes in solid-state dye-sensitized solar cells: Towards optically-active metal oxides for photovoltaic applications. Solar Energy Materials and Solar Cells 2013, 117, 624-631.
  12. Wang, W.; Liu, Y.; Sun, J.; Gao, L., Nitrogen and yttrium co-doped mesoporous titania photoanodes applied in DSSCs. Journal of Alloys and Compounds 2016, 659, 15-22.
  13. Jo, I.-R.; Lee, Y.-H.; Kim, H.; Ahn, K.-S., Multifunctional nitrogen-doped graphene quantum dots incorporated into mesoporous TiO2 films for quantum dot-sensitized solar cells. Journal of Alloys and Compounds 2021, 870, 159527.
  14. Kim, S.-B.; Park, J.-Y.; Kim, C.-S.; Okuyama, K.; Lee, S.-E.; Jang, H.-D.; Kim, T.-O., Effects of graphene in dye-sensitized solar cells based on nitrogen-doped TiO2 composite. The Journal of Physical Chemistry C 2015, 119, (29), 16552-16559.
  15. Nguyen, P. T.; Andersen, A. R.; Skou, E. M.; Lund, T., Dye stability and performances of dye-sensitized solar cells with different nitrogen additives at elevated temperatures—Can sterically hindered pyridines prevent dye degradation? Solar energy materials and solar cells 2010, 94, (10), 1582-1590.
  16. Xie, Y.; Huang, N.; Liu, Y.; Sun, W.; Mehnane, H. F.; You, S.; Wang, L.; Liu, W.; Guo, S.; Zhao, X.-Z., Photoelectrodes modification by N doping for dye-sensitized solar cells. Electrochimica Acta 2013, 93, 202-206.
  17. Ngwenya, S. N. Application of N-doped TiO₂/Bi₂O₃/rGO nanocomposites as a photoanode material in dye-sensitized solar cells. 2021.
  18. Guo, W.; Shen, Y.; Boschloo, G.; Hagfeldt, A.; Ma, T., Influence of nitrogen dopants on N-doped TiO2 electrodes and their applications in dye-sensitized solar cells. Electrochimica Acta 2011, 56, (12), 4611-4617.
  19. Lim, S. P.; Pandikumar, A.; Huang, N. M.; Lim, H. N.; Gu, G.; Ma, T. L., Promotional effect of silver nanoparticles on the performance of N-doped TiO 2 photoanode-based dye-sensitized solar cells. Rsc Advances 2014, 4, (89), 48236-48244.
  20. Park, J.-Y.; Kim, C.-S.; Okuyama, K.; Lee, H.-M.; Jang, H.-D.; Lee, S.-E.; Kim, T.-O., Copper and nitrogen doping on TiO2 photoelectrodes and their functions in dye-sensitized solar cells. Journal of Power Sources 2016, 306, 764-771.
  21. Qin, W.; Lu, S.; Wu, X.; Wang, S., Dye-sensitized solar cell based on N-doped TiO2 electrodes prepared on titanium. Int. J. Electrochem. Sci 2013, 8, (6), 7984-7990.
  22. Diker, H.; Varlikli, C.; Stathatos, E., N‐doped titania powders prepared by different nitrogen sources and their application in quasi‐solid state dye‐sensitized solar cells. International journal of energy research 2014, 38, (7), 908-917.
  23. Motlak, M.; Akhtar, M. S.; Barakat, N. A.; Hamza, A.; Yang, O.-B.; Kim, H. Y., High-efficiency electrode based on nitrogen-doped TiO2 nanofibers for dye-sensitized solar cells. Electrochimica Acta 2014, 115, 493-498.
  24. Tran, V. A.; Truong, T. T.; Phan, T. A. P.; Nguyen, T. N.; Van Huynh, T.; Agresti, A.; Pescetelli, S.; Le, T. K.; Di Carlo, A.; Lund, T., Application of nitrogen-doped TiO2 nano-tubes in dye-sensitized solar cells. Applied Surface Science 2017, 399, 515-522.
  25. Duarte, D.; Sagás, J.; da Silva Sobrinho, A.; Massi, M., Modeling the reactive sputter deposition of N-doped TiO2 for application in dye-sensitized solar cells: effect of the O2 flow rate on the substitutional N concentration. Applied Surface Science 2013, 269, 55-59.
  26. Rajaramanan, T.; Kumara, G.; Velauthapillai, D.; Ravirajan, P.; Senthilnanthanan, M., Ni/N co-doped P25 TiO2 photoelectrodes for efficient Dye-Sensitized Solar Cells. Materials Science in Semiconductor Processing 2021, 135, 106062.
  27. Mousa, M.; Khairy, M.; Mohamed, H., Dye-sensitized solar cells based on an N-doped TiO2 and TiO2-graphene composite electrode. Journal of Electronic Materials 2018, 47, (10), 6241-6250.
  28. Li, L.; Wang, C.-L.; Liao, J.-Y.; Manthiram, A., Dual-template synthesis of N-doped macro/mesoporous carbon with an open-pore structure as a metal-free catalyst for dye-sensitized solar cells. Journal of Power Sources 2015, 300, 254-260.

Round 2

Reviewer 2 Report

The authors addressed all the concerns and improved the manuscript. Two small aspects to be considered: first, with respect to the XPS, authors could mention that there were some contamination due to processing of the sample (typically some contamination with C always occurs), and second, for the N1s fit, the FWHM of the N2 absorbed peak is very low. To note, that typically fitted peaks have similar range (+- 0.2-0.3eV) of the FWHM and not such a high difference. Authors could search for additional references for this fitted N peak to further confirm.

Please check manuscript one more time for small errors/misspellings .

Author Response

AUTHOR’S RESPONSE TO REVIEWER # 2 COMMENTS:

< materials-1855674R2>

< Electrochemical impedance investigation of dye-sensitized solar cells based on electrospun TiO2 nanofibers photoanodes >

Dear Editor of “Materials”

Firstly, we would like to thank you for giving us a chance to resubmit the work after revision, and also thank the reviewers for giving us constructive suggestions which would help us to improve the quality of the paper. Here we submit a revised version of our manuscript, which has been modified according to the editor and reviewers’ suggestions. We mark the changes in yellow color in the new version manuscript. The detailed corrections are listed below point by point:  

RESPONSE TO REVIEWER # 2 COMMENTS: 

The authors addressed all the concerns and improved the manuscript. Two small aspects to be considered: first, with respect to the XPS, authors could mention that there were some contamination due to processing of the sample (typically some contamination with C always occurs), and second, for the N1s fit, the FWHM of the N2 absorbed peak is very low. To note, that typically fitted peaks have similar range (+- 0.2-0.3eV) of the FWHM and not such a high difference. Authors could search for additional references for this fitted N peak to further confirm.

Please check manuscript one more time for small errors/misspellings .

We would like to thank the reviewer for his great efforts and for giving an important criticism of the article.

Author reply: Thanks for the reviewer’s comment. XPS was used in this study to investigate the surface chemistry of the prepared TiO2 NFs (Fig. 4A). From XPS data, the peaks assigned to O 1s [1], N 1s [2], and Ti 2p [3] could be seen at 529.1 eV, 398.9 eV, and 457.9 eV, respectively. These peaks indicate the existence of TiO2 and provided acceptable proof for the successful N- doping by the XPS peak of N 1s at 398.9 eV [2]. The nitrogen peak was deeply studied at the fine spectrum as shown in Fig. 4B. Two clear deconvoluted peaks could be seen at 399.07 eV, and 398.48 eV which could be assigned to different N-bonds and expected to be related to the substitutional N instead of O-atoms in the anatase crystal of TiO2 NFs, and N atoms inside the interstitial locations between anatase crystals to form Ti–O–N [4]. According to Koli et al., the major peak located at 399.07 eV (higher BE) could be attributed to the substitutional N-atoms inside the TiO2 crystals, and a second peak at 398.48 eV could be due to the N-atoms in the interstitial locations to form Ti–O–N bond. The other small narrow peak has no high impact on the discussion and conclusions regarding the larger peaks discussed above. The spectra could have been fitted without this narrow peak, but the overall quality of the fit and the consistency between the experimental peak and peak summation should be exactly the same as much as we can. For this small peak, it could be due to another type of nitrogen at the depth of 10 nm which is the trapping N2-molecules inside the anatase crystals of TiO2 [5, 6]. In short, the XPS analysis indicates the surface chemistry of the synthesized N-TiO2 NFs to be TiO2 and doped by nitrogen atoms.

The manuscript was revised and checked to be free from any small errors or misspelling.

Thank you for your efforts and comments, we really find the comments valid and useful.

I hope that all modifications have taken into account the suggestions of the editor and reviewers. The manuscript has been resubmitted to your journal. We look forward to your positive response.   

Sincerely,  

Prof. Hany M. Abd El-Lateef

REFERENCES

  1. Zhang, J.; Raza, S.; Wang, P.; Wen, H.; Zhu, Z.; Huang, W.; Mohamed, I. M. A.; Liu, C., Polymer brush-grafted ZnO-modified cotton for efficient oil/water separation with abrasion/acid/alkali resistance and temperature “switch” property. Journal of Colloid and Interface Science 2020, 580, 822-833.
  2. Iqbal, W.; Yang, B.; Zhao, X.; Rauf, M.; Mohamed, I. M. A.; Zhang, J.; Mao, Y., Facile one-pot synthesis of mesoporous g-C3N4 nanosheets with simultaneous iodine doping and N-vacancies for efficient visible-light-driven H2 evolution performance. Catalysis Science & Technology 2020, 10, (2), 549-559.
  3. Mohamed, I. M. A.; Dao, V.-D.; Yasin, A. S.; Mousa, H. M.; Mohamed, H. O.; Choi, H.-S.; Hassan, M. K.; Barakat, N. A. M., Nitrogen-doped&SnO2-incoportaed TiO2 nanofibers as novel and effective photoanode for enhanced efficiency dye-sensitized solar cells. Chemical Engineering Journal 2016, 304, 48-60.
  4. Koli, V. B.; Mavengere, S.; Kim, J.-S., An efficient one-pot N doped TiO2-SiO2 synthesis and its application for photocatalytic concrete. Applied Surface Science 2019, 491, 60-66.
  5. Shah, D.; Bahr, S.; Dietrich, P.; Meyer, M.; Thißen, A.; Linford, M. R., Nitrogen gas (N2), by near-ambient pressure XPS. Surface Science Spectra 2019, 26, (1), 014023.
  6. Kusunoki, I.; Sakai, M.; Igari, Y.; Ishidzuka, S.; Takami, T.; Takaoka, T.; Nishitani-Gamo, M.; Ando, T., XPS study of nitridation of diamond and graphite with a nitrogen ion beam. Surface Science 2001, 492, (3), 315-328.

Reviewer 3 Report

Accept

Author Response

AUTHOR’S RESPONSE TO REVIEWER # 3 COMMENTS:

< materials-1855674R2>

< Electrochemical impedance investigation of dye-sensitized so-lar cells based on electrospun TiO2 nanofibers photoanodes >

Dear Editor of “Materials”

Firstly, we would like to thank you for giving us a chance to resubmit the work after revision, and also thank the reviewers for giving us constructive suggestions which would help us to improve the quality of the paper. Here we submit a revised version of our manuscript, which has been modified according to the editor and reviewers’ suggestions. We mark the changes in yellow color in the new version manuscript  

RESPONSE TO REVIEWER # 3 COMMENTS: 

We would like to thank the reviewer for his great efforts and for giving an important criticism of the article.

Thank you for your efforts and comments, we really find the comments valid and useful.

I hope that all modifications have taken into account the suggestions of the editor and reviewers. The manuscript has been resubmitted to your journal. We look forward to your positive response.   

Sincerely,  

Prof. Hany M. Abd El-Lateef
